# Self-contrastive weakly supervised learning framework for prognostic prediction using whole slide images

**Saul Fuster**[1�*], **Farbod Khoraminia**[2౿], **Julio Silva-Rodríguez**[3], **Umay Kiraz**[4,5], **Geert J. L. H. van Leenders**[6], **Trygve Eftestøl**[1], **Valery Naranjo**[7], **Emiel A. M. Janssen**[4,5], **Tahlita C. M. Zuiverloon**[2], **Kjersti Engan**[1*]

1 Department of Electrical Engineering and Computer Science, University of Stavanger, Stavanger, Norway, 2 Department of Urology, Erasmus MC Cancer Institute, University Medical Center, Rotterdam, The Netherlands, 3 ÉTS Montréal, Montréal, Québec, Canada, 4 Department of Pathology, Stavanger University Hospital, Stavanger, Norway, 5 Department of Chemistry, Bioscience and Environmental Engineering, University of Stavanger, Stavanger, Norway, 6 Department of Pathology and Clinical Bioinformatics, Erasmus MC Cancer Institute, University Medical Center Rotterdam, Rotterdam, The Netherlands, 7 CVBLab, Instituto Universitario de Investigación en Tecnología Centrada en el Ser Humano (HUMAN-tech), Universitat Politècnica de València, Valencia, Spain

౿ These authors contributed equally to this work.
* saul.fusternavarro@uis.no (SF); kjersti.engan@uis.no (KE)

**Data availability statement:** Data cannot be shared publicly since data contain potentially

## Abstract

We present a pioneering investigation into the application of deep learning techniques to analyze histopathological images for addressing the substantial challenge of automated prognostic prediction. Prognostic prediction poses a unique challenge as the ground truth labels are inherently weak, and the model must anticipate future events that are not directly observable in the image. To address this challenge, we propose a novel three-part framework comprising of a convolutional network based tissue segmentation algorithm for region of interest delineation, a contrastive learning module for feature extraction, and a nested multiple instance learning classification module. Our study explores the significance of various regions of interest within the histopathological slides and exploits diverse learning methods in real-world clinical scenarios. The pipeline is initially validated on artificially generated data and a simpler diagnostic task. Transitioning to prognostic prediction, tasks become more challenging. Employing bladder cancer as use case, our best models yield an AUC of 0.721 and 0.678 for recurrence and treatment outcome prediction respectively for a private data cohort. Altogether, this research serves as an initial investigation on the shortcomings of histopathological image analysis for treatment outcome prediction.

## Author summary

Predicting disease outcomes, or prognostic prediction, is more challenging than diagnosis due to factors like tissue variability and treatment response diversity.

identifying or sensitive patient information. Data are available from the Regional Committees for Medical and Health Research Ethics (REC), Norway, ref.no.: 2011/1539, regulated in accordance to the Norwegian Health Research Act; and the Daily Board of the Medical Ethics Committee Erasmus MC, Rotterdam, The Netherlands, METC number: MEC-2019-704; for researchers who meet the criteria for access to confidential data. Inquiries regarding data availability should be addressed at patologi@sus.no and blaaskankercentum@erasusmc.nl.

**Funding:** This research has received funding from the European Union's Horizon 2020 research and innovation program under grant agreements 860627 (CLARIFY). The funders had no role in study design, data collection and analysis, decision to publish, or preparation of the manuscript.

**Competing interests:** The authors have declared that no competing interests exist.

Urinary bladder cancer prognosis, in particular, is complicated by heterogeneous behaviors and lacking clear markers. While diagnostic research is prevalent in computational pathology, prognostic prediction receives less attention due to data complexities. Our study introduces an automated method leveraging deep learning to predict urinary bladder cancer prognosis on large unlabelled data. By focusing on key tissue areas and utilizing advanced machine learning and image processing techniques, our approach enhances prognostic accuracy. While not without limitations, this work marks a significant advancement in urinary bladder cancer prognosis, laying the foundation for future research.

## Introduction

The introduction of digital pathology, characterized by the digitization of tissue sections into whole slide images (WSI) through microscopy scanners, has opened up numerous possibilities. A prevailing trend in computational pathology (CPATH) research involves utilizing image processing and machine learning to develop tools that assist pathologists in visualization, region of interest (ROI) extraction, and diagnostic tasks [1].

Prognostic prediction is generally acknowledged as more challenging than diagnostic prediction as it involves forecasting future events and outcomes. Prognosis is arduous due to histological diversity, observer variability, and tumor heterogeneity [2]. Additionally, diverse treatment responses and recurrence patterns must be considered. Urinary bladder prognostic prediction exemplifies this complexity as a result of diverse treatment responses, recurrence patterns, and the absence of distinct markers and varied clinical factors [3]. Additionally, timely interventions can greatly enhance treatment efficacy and overall prognosis, making it essential for clinicians to make informed decisions based on the initial tumor observation.

Plenty of research in CPATH is dedicated to diagnostic prediction, often relying on manually selected ROIs during both model learning and inference stages. For prognostics, weakly labeled data is commonly employed, with patient-based labels defining treatment outcomes, recurrence, or disease progression. Consequently, there is no guarantee that a particular region contains the necessary information, or that the essential information is genuinely present in the WSI.

In this study, we present an automated deep learning pipeline for prognostic predictions from WSI, trained and tested on weakly labeled data from a private cohort. A nested multiple instance learning method with attention mechanisms (NMIA), recently proposed by our research group, is used for the first time in an end-to-end problem in a real-world clinical setting [4]. The pipeline utilizes extensive unannotated regions to enhance feature representations and explores the impact of selectively choosing informative areas within the slides. This work represents a pioneering investigation into the application of prognostic predictions in urinary bladder cancer. The main contributions of this paper are summarized as:

1. Introduction of an automated deep learning pipeline for prognostic predictions from WSI, leveraging weakly labeled data. The pipeline incorporates NMIA, a novel approach applied for the first time in an end-to-end problem.
2. Utilization of extensive unannotated regions to enrich feature representations and investigation into the impact of selectively choosing informative areas within the slides guided by domain knowledge, marking a pioneering exploration into the application of prognostic predictions in urinary bladder cancer.

## Urinary bladder cancer

Non-muscle invasive bladder cancer (NMIBC) conform approximately 75% of bladder cancer diagnoses [5]. The 2022 version of the European Association of Urology (EAU) guidelines on NMIBC suggests that patients are stratified into risk groups based on the hazard to progress to a muscle-invasive disease [3]. The hazard score depends on significant clinical and pathological factors, which are themselves time-consuming to determine and frequently result in variations among uropathologists. Currently, intravesical Bacillus Calmette–Guérin (BCG) is the gold standard adjuvant therapy for high-risk non-muscle invasive bladder cancer (HR-NMIBC). Unfortunately, BCG treatment causes many severe adverse reactions, and up to 50% of the patients will develop BCG resistance, thus resulting in recurrence or progression [6]. Identifying these patients at the first trans urethral resection of bladder tumor (TURBT) would significantly reduce the recurrence rate and treatment cost, hence contributing to more adequate patient-based treatment strategies. Nonetheless, bladder WSI are sizable and often contain disorganized, fragmented tissue sections, with a significant number of artifacts and other non-diagnostically-relevant tissue [5]. The unique challenges of urinary bladder cancer WSI have served as inspiration for the three-step pipeline presented in this study. Also, to the best of our knowledge, a deep learning system for BCG response prediction (BCG-RP) relying on image features does not exist in the literature.

## Data modalities

Computer-aided diagnosis (CAD) systems that utilize machine learning techniques for medical imaging analysis of WSI have shown effective ways to reduce subjectivity and speed up the diagnostic process [1]. WSI are pre-stored at various magnification levels, allowing pathologists to quickly adjust the zoom level, analogous to physical microscopes. Lower magnification is typically used to view tissue-level morphology, while higher magnification is useful for examining cell-level features. In CPATH, imaging techniques that rely on convolutional neural networks (CNNs) are considered the best option for extracting features from histological images [7]. Traditionally, imaging has been crucial for diagnostics and risk stratification due to its ability to clearly observe and measure features directly related to the state of the tumor. In contrast, prognostics often lack a direct correlation between observable features and clinical outcomes, making prediction based on imaging alone more challenging [8]. Consequently, prognostic applications have primarily relied on clinicopathological information more than on image data [9,10]. Both clinicopathological information and image features are derived from the visual characteristics of tumor tissue. Image features are extracted directly from the image, while clinicopathological information depends on external observations. Nonetheless, recent deep learning prognostic applications employ image features, while providing reasonably accurate prognostic predictions. Although there are methods based on clinicopathological or image data alone, hybrid solutions have also been proposed with state-of-the-art performance [11,12]. Another promising type of data is that of genomics, specifically next-generation sequencing, which has altered the understanding and assessment of cancer [13]. However, next-generation sequencing is an expensive technology, still in its early implementation phase, making it an inaccessible solution for many pathology laboratories at present. In contrast to this, hematoxylin and eosin-stained (H&E) WSI provide an affordable solution and practical choice for routine diagnostic and research purposes. Therefore, we explore H&E WSI both with and without clinical information in this study.

### Non-supervised learning methods

One challenge in CPATH is the lack of WSI with detailed or region of interest (ROI) annotations, which is often expensive and time-consuming to acquire. To address this, weakly supervised methods have been proposed for training deep learning models on WSI [1,14–17]. Weakly supervised methods emerge as an advantageous approach for prognostic applications as they accommodate the uncertainty and heterogeneity of medical data [18,19]. Among the diverse weakly supervised methods, attention-based multiple instance learning (AbMIL) is a popular approach [20–23]. The method uses an attention mechanism to selectively focus on regions of interest within an image, allowing the model to learn from weakly labeled data. One diagnostic application using weakly supervised deep learning on WSI is that of predicting the pathological grade of the patient [24–26]. Such approach was able to achieve performance on par with supervised methods, while reducing the amount of annotated data required. Cancer survival prediction using WSI was performed with AbMIL, as demonstrated in [27–31]. It is highlighted that the attention-based approach improved the performance of the model. However, WSI present scattered tissue, with countless instances. Hence, a straightforward MIL approach may not be suitable, as WSI could be densely populated with noisy instances. This led us to propose NMIA, which restricts cross-contamination among regions within the images [4].

The relationship between image features and patient outcomes can be complex and difficult to discern. To address this lack of correlation, self-supervised methods can overcome the disparity. In recent years, contrastive learning has emerged as a promising technique for learning feature representation from large unlabeled datasets. Contrastive learning is a type of learning with the aim of training a feature extractor, using a contrastive loss function [32]. This learning approach has the capacity to utilize extensive unannotated regions for enhancing feature representation. Typically, a contrastive module serves as a preliminary step before classification, facilitating the extraction of feature representations [33,34]. Moreover, the acquisition of transformation-independent features has proven effective in mitigating stain variation [35,36]. It has also been employed for maximizing feature similarity between areas from the same WSI [37,38]. Contrastive learning in WSI prognostics is mostly unexplored [39]. We adopt SimCLR and variations for exploiting underlying prognostic patterns in WSI [32].

## Methods

First, we define the dataset utilized to undergo this study. Then, in the subsequent subsections, we introduce a three-step fully-automated pipeline for WSI prognosis that combines region of interest (ROI) extraction, contrastive learning for feature representation and multiple instance learning (MIL) for predicting the concluding prognostic outcome, as depicted in Fig 1. This approach enables us to optimize the model by leveraging the benefits of these techniques. It ensures the inclusion of important instances for predicting clinical outcome from WSI visual cues, while maintaining computational feasibility. Ultimately, the proposed steps for prognostic predictions in histopathological imaging are the following:

1. Define and extract ROIs using a tissue segmentation algorithm for tile extraction strategies.
2. Train a feature extractor $G_\theta$ to generate an intermediate dataset $\mathcal{H}$ using contrastive learning.
3. Use image feature embeddings $\mathcal{H}$ for prognostic classification using MIL.

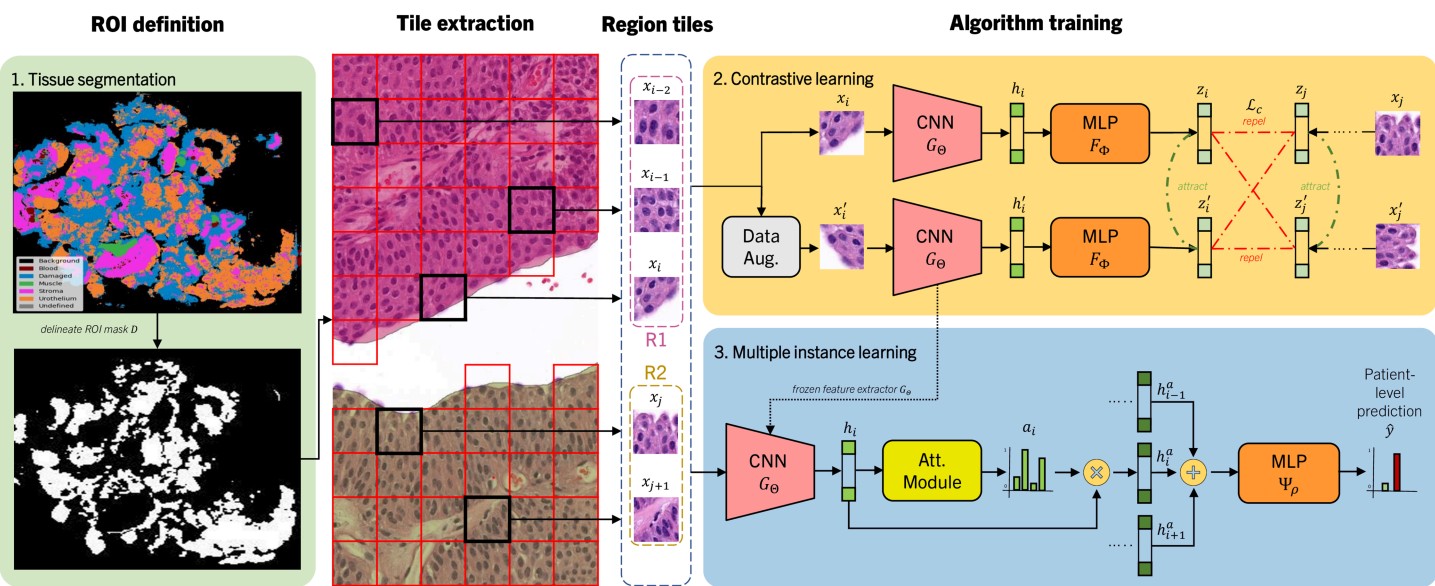

**Fig 1. Deep learning pipeline for prognostic outcome prediction.** 1) A tissue segmentation is employed for delineating a ROI of choice $D$. Then,tiles are extracted from WSI regions for training an algorithm. 2) Contrastive learning is employed to learn representations of the tiles. 3) The representations are then used to train an AbMIL model that predicts the prognostic outcome. This approach compresses an end-to-end pipeline where the raw input image data is broken down and processed for predicting clinical outcome.

## Dataset

In this study, we have gathered NMIBC WSI from two distinct cohorts. The total number of patients per application is displayed in Table 1. For either cohort, a comprehensive quality assessment of the visual features present in the WSIs was conducted by expert uropathologists. Additionally, it was confirmed that the clinical data documented in the records were accurately represented in the slides. To address potential biases in the study population, a meticulous data curation process was performed, and a data split was established that considered clinical variables deemed representative by expert uropathologists. Finally, to the best of our knowledge, no NMIBC WSI datasets are currently publicly available.

We have available HR-NMIBC WSI from an initial TURBT from a multi-centre cohort provided by Erasmus Medical Center (EMC), Rotterdam, The Netherlands. We denote this dataset $S_{EMC}$, and use it for BCG-RP. Let BCG-R and BCG-NR denote BCG responder and non-responder tumor, correspondingly. BCG-NR corresponds to BCG failure according to EAU guidelines, excluding BCG intolerance. A total of 453 patients and 503 WSI formed the dataset. Since the treatment outcome is related to the patient as a whole and not to a specific

**Table 1. Description of the patient sets, $S_{EMC}$ and $S_{SUH}$, and how many patients in each group divided in train/val/test splits.**

| Set | $S_{EMC}$ | | $S_{SUH}$ | |
|---|---|---|---|---|
| | **BCG-R** | **BCG-NR** | **Rec** | **NoRec** |
| Train | 272 (72) | 81 (42) | 113 (0) | 107 (0) |
| Validation | 25 (24) | 25 (22) | 18 (0) | 12 (0) |
| Test | 25 (25) | 25 (25) | 27 (0) | 23 (0) |

*The number between parenthesis indicates the number of patients with annotated WSI.*

region of the tumor, patches from various WSI that belong to the same patient were merged as a single entity. Not all slides contained annotations, and among those that were annotated, none of the WSI were fully annotated due to time and database storage constraints. Annotations contain tissue types, artifacts, grading and staging. Moreover, a detailed report of clinicopathological information per patient was disclosed with information about BCG treatment guidelines, pathological diagnoses, and patient demographics. We utilized clinical variables of gender, age, smoking status, grade, stage, concomitant carcinoma in situ, size and focality of the tumor. See Support Information S1 Table for further details on variable values.

We also included a total of 300 WSI corresponding to 300 different NMIBC patients from Stavanger University Hospital (SUH), Stavanger, Norway. We denote this dataset $S_{SUH}$, and use it for recurrence prediction. Let NoRec and Rec denote no recurrence and recurrence, respectively. Rec was defined as recurrent tumors in the bladder only, with a median follow-up of 82 months. No WSI were annotated with ROI, but weak labels regarding recurrence outcome were available.

With regards to ROI definition, two main strategies were employed: using annotated areas or areas defined from an automatic tissue segmentation algorithm. Fig 2 highlights the various ROIs explored in this study. The automatic definition of ROIs is later described in Automatic Region of Interest Segmentation. Concerning annotations, an engineer with specific training in bladder pathology (F. K.) partially annotated 217 of the total 503 EMC patient slides from $S_{EMC}$, under the supervision of an experienced uropathologist (G. vL.). This subset of data is referred as $D_{ANNO}$. The annotation process consisted of general tissue type annotations, and in some cases, sub classes indicating grading, presence of tumor infiltrating lymphocytes, flat lesions, and invasive areas. First, a coarse annotation of the WSI was done, identifying and labeling the main regions of interest within the images. Then, a quality control process was implemented to ensure the annotation's quality and consistency by getting external revision from expert uropathologists.

**Compliance with ethical standards.** This work involved human subjects in its research. Approval of all ethical and experimental procedures and protocols was granted by the Regional Committees for Medical and Health Research Ethics (REC), Norway, ref.no.: 2011/1539, regulated in accordance to the Norwegian Health Research Act; and the Daily Board of the Medical Ethics Committee Erasmus MC, Rotterdam, The Netherlands, METC number: MEC-2019-704. Inquiries regarding data availability should be addressed at `patologi@sus.no` and `blaaskankercentum@erasusmc.nl`.

## Automatic region of interest segmentation

The current study will explore various ROI configurations as the localization of the tissue of interest is unknown. While annotations can be expensive and inflexible, automatic tissue segmentation algorithms provide the flexibility to redefine ROIs based on different clinical considerations. A tissue segmentation algorithm was proposed in [40] for $S_{SUH}$, while an active learning-based approach for tissue segmentation was developed in [41] for $S_{EMC}$. Both models share the same architecture, utilizing a tri-scale CNN backbone that leverages different magnifications for each input CNN. The tissue segmentation algorithms work at patch level, and classify all patches $x$ in the WSI as $y \in \mathcal{Y}$ = {urothelium, lamina propria, muscle, blood, damage, background}. The dataset resulting from extracting patches with label $y$ is denoted $D_y$. For example, urothelium tissue is the most prominent source of information in urothelial bladder carcinoma, and the dataset of patches extracted from these regions are denoted $D_{URO}$. In conjunction with urothelium, lamina propria may serve an important role for influencing the growth of the tumor [42]. The dataset of patches extracted from lamina propria is

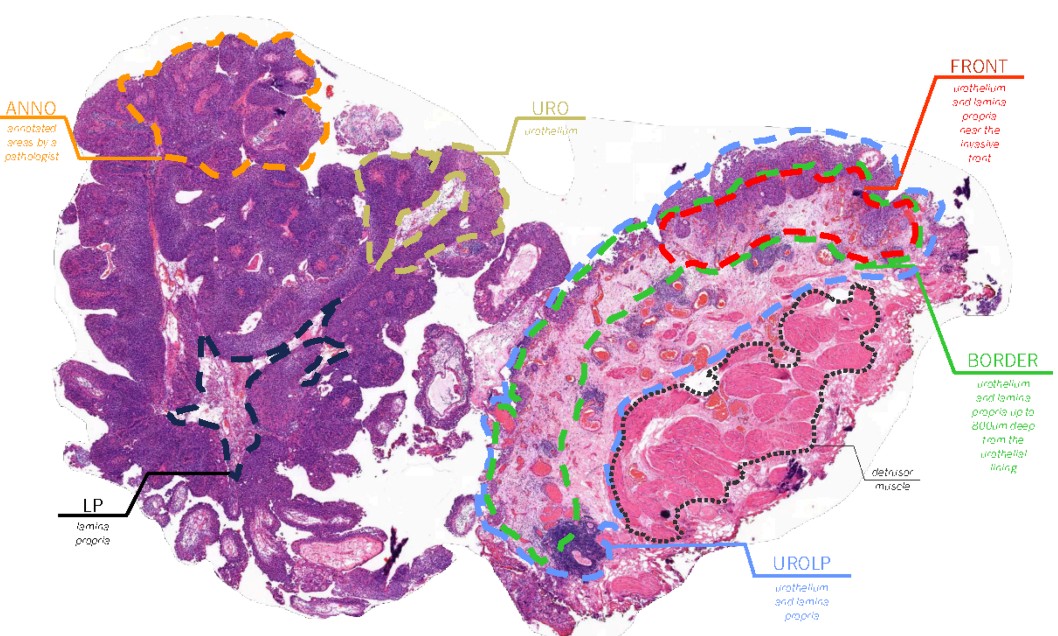

**Fig 2. Overview of ROIs utilized.** The process of extracting ROIs from raw WSI involved either a tissue segmentation algorithm and/or pathologist's annotations. The annotations highlighted areas that were deemed prognostically significant for predicting outcome, while the algorithm provided masks that highlighted different tissue types. Subsets of tissue were extracted using urothelium and lamina propria. For magnification levels, the study explored two mono-scale approaches using 10x and 20x, as well as a multi-scale method using three magnifications (2.5x, 10x, 40x).

denoted $D_{LP}$, and the union urothelium and lamina propria $D_{UROLP} = D_{URO} \cup D_{LP}$. However, these $D_y$ are solely defined based on tissue types, and to meticulously analyze and concentrate on the pertinent area of interest for comprehending the disease's status, it is imperative to define tailored ROIs. Consequently, exploiting domain knowledge through segmentation maps is a pivotal aspect of this work. Notably, not all lamina propria might be interacting with the tumor. Therefore, we defined a depth of $800\mu m$ based on medical knowledge for defining possible tumor and immune response interactions based on medical knowledge. The union of the boundary between urothelium and lamina propria defines the dataset $D_{BORDER} \subset D_{UROLP}$. This is accomplished by applying a disk dilation operation on the urothelium and lamina propria masks, where the disk radius is determined by pixel size, and subsequently segment the overlapping area to extract the bordering region. Fig 3 displays an schematic representation of the self-defined ROI $D_{BORDER}$. Furthermore, the invasive front of the tumor, which represents the most aggressive part of the tumor, could potentially provide the most significant features for comprehending the current state of the tumor in relation to the patient's immune system [43]. Therefore, we refine $D_{BORDER}$ using a region-growing algorithm along these borders to exclude areas lacking muscle tissue within a tissue section, thus defining $D_{FRONT} \subset D_{BORDER}$. To exclude distant muscle areas from consideration, we apply the same distance threshold of $800\mu m$, thus ensuring the focus remains on the invasive front regions.

Regarding tile extraction strategies, various magnification levels and tile sizes are used. In the case of mono-scale models, we used a tile size of $256 \times 256$ for 10x magnification and $512 \times 512$ for 20x magnification. This decision was made to ensure that the patches covered the same physical area, i.e. field of view. In the case of multi-scale models TRI, we used a tile

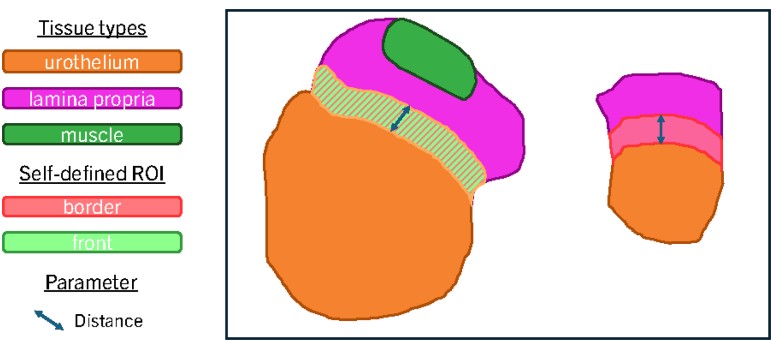

**Fig 3. Schematic representation of self-defined ROI generation.** Guided by the segmentation mask of various tissue types, we apply diverse image processing morphological operations to define ROIs $D_y$ based on domain knowledge from expert pathologists. In the example displayed, we enlarge the urothelium and lamina propria masks applying dilation, limited by a distance parameter determined on clinical expertise. The resulting overlapping areas represent $D_{\text{BORDER}}$. We further delineate a subset of $D_{\text{BORDER}}$ by extracting only those areas where muscle is present within the same tissue section, aiming to represent the potential invasive front in $D_{\text{FRONT}}$.

size of $128 \times 128$ for all three magnification levels (40x, 10x, and 2.5x), extracted following the approach described in [44].

## Feature extraction via contrastive learning

Contrastive learning is a specific approach within metric learning that focuses on comparing similar and dissimilar pairs of examples, based on a siamese structure, by attracting and repelling representations accordingly [32]. A contrastive model is trained as follows: given a batch of $N$ random samples from a training set of $X$ images, a set of two image transformations are applied to all images in the batch in order to obtain $2N$ augmentations. These transformed images are forwarded though a feature extractor $G_\theta : \mathcal{X} \to \mathcal{H}$ and projected into a low-dimensional feature space through a multi-layer perceptron (MLP) $F_\phi : \mathcal{H} \to \mathcal{Z}$, to be later $l_2$ normalized. Let $\text{sim}(\mathbf{z}_i, \mathbf{z}_j) = \mathbf{z}_i^\top \mathbf{z}_j / \|\mathbf{z}_i\| \|\mathbf{z}_j\|$ be the cosine similarity between $l_2$ normalized vectors $\mathbf{z}_i$ and $\mathbf{z}_j$, then the loss function for a positive pair is defined as:

$$\mathcal{L}_c = -\frac{1}{2N} \sum_{i \in I} \log \frac{\exp(\text{sim}(\mathbf{z}_i, \mathbf{z}_i')/\tau)}{\sum_{j=1}^{2N} 1_{[j \neq i]} \exp(\text{sim}(\mathbf{z}_i, \mathbf{z}_j)/\tau)} \tag{1}$$

where $\mathbf{z}_i'$ is the augmented representation of $\mathbf{z}_i$, $1_{[j \neq i]}$ is a binary indicator to indicate all other instances than $i$, and $\tau$ is a temperature coefficient to control the strenght of penalties on hard negative samples. The contrastive loss has the objective of ensuring the model learns strong feature representations regardless of the augmentation applied, thus increasing the robustness to variability in the input image. The contrastive loss rewards the model for creating similar features for both augmentations from the same image, while increasing feature dissimilarity between other augmentations from images in the batch. $\mathcal{L}_c$ corresponds to the unsupervised version, although the supervised contrastive learning loss $\mathcal{L}_{sc}$ does exist [45]. The supervised loss $\mathcal{L}_{sc}$ considers images from the same class in the batch, using the corresponding image label $y$, and does not punish the model for generating similar representations among images

from the same class:

$$\mathcal{L}_{sc} = \sum_{i \in I} \frac{-1}{|P_{(i)}|} \sum_{p \in P_{(i)}} \log \frac{\exp(\mathrm{sim}(\mathbf{z_i}, \mathbf{z_i}')/\tau)}{\sum_{j=1}^{2N} 1_{[j \neq i]} \exp(\mathrm{sim}(\mathbf{z_i}, \mathbf{z_j})/\tau)} \quad (2)$$

where $P_{(i)} \equiv p : y_p = y_i$, while $|P_{(i)}|$ represents its cardinality. $\mathcal{L}_{sc}$ is preferred when labeled data is available, allowing for explicit learning of relevant representations and improved model performance. However, when labeled data is scarce or difficult to obtain, $\mathcal{L}_c$ can be a practical option.

We further explore the implementation of contrastive learning defining a multi-task learning loss. For multi-task learning, two separate projection heads for both $\mathcal{L}_c$ and cross entropy loss $\mathcal{L}_{ce}$ are simultaneously trained. $\mathcal{L}_{ce}$ is calculated using the output predictions of a classifier module $C_\eta : \mathcal{H} \to \mathcal{Y}$. As labels are a requirement for calculating $\mathcal{L}_{ce}$, $D_{\mathrm{ANNO}}$ is employed. For computing the loss for multi-task contrastive learning, we combine the loss from two separate projections:

$$\mathcal{L}_{multi} = \alpha_c \mathcal{L}_c + \alpha_{ce} \mathcal{L}_{ce} \quad (3)$$

where $\alpha_c$ and $\alpha_{ce}$ are the scaling factors for the unsupervised contrastive and supervised cross entropy losses, respectively.

## Prognostic outcome classification via multiple instance learning

Multiple instance learning (MIL) is a suitable approach for prognostic classification due to its ability to handle inherent uncertainties, diversity and intricacies present in medical data [46]. A dataset $\mathcal{H}, \mathcal{Y} = \left\{ (\mathbf{H}^i, y^i), \forall i = 1, ..., N \right\}$ is formed of pairs of bag instances $\mathbf{H}$ and their corresponding labels $y$, where $i$ denotes the current sample for a total of $N$ samples. A bag $\mathbf{H}$ consists of instances $\mathbf{h}_l$:

$$\mathbf{H} = \left\{ \mathbf{h}_l, \forall l = 1, ..., L \right\} \quad (4)$$

where $L$ is the number of instances in the bag. Among MIL model architecture variants, we adopted attention-based multiple instance learning (AbMIL). Given a label for a patient, a model should infer which ROIs visual features lead to predicting the patient's prognostic outcome. An attention score $a_i$ for a feature embedding $\mathbf{h}_i$ can be calculated as:

$$a_i = \frac{\exp\{\mathbf{w}^\top (\tanh(\mathbf{V}\mathbf{h_i}^\top) \odot \mathrm{sigm}(\mathbf{U}\mathbf{h_i}^\top))\}}{\sum_{l=1}^{L} \exp\{\mathbf{w}^\top (\tanh(\mathbf{V}\mathbf{h_l}^\top) \odot \mathrm{sigm}(\mathbf{U}\mathbf{h_l}^\top))\}} \quad (5)$$

where $\mathbf{w} \in \mathbb{R}^{L \times 1}$, $\mathbf{V} \in \mathbb{R}^{L \times M}$ and $\mathbf{U} \in \mathbb{R}^{L \times M}$ are trainable parameters and $\odot$ is an element-wise multiplication. Furthermore, the hyperbolic tangent $\tanh(\cdot)$ and sigmoid $\mathrm{sigm}(\cdot)$ are included to introduce non-linearity for learning complex applications. The strength of the attention modules is not only in terms of interpretability, but also in predictive power, as attention scores are directly influencing the forward propagation of the model. Once the attention scores are obtained, we obtain the patient prediction $\hat{y}$ as:

$$\hat{y} = \Psi_\rho (\mathbf{A} \cdot \mathbf{H}) \quad (6)$$

where $\Psi_\rho$ is a MLP acting as a patient classifier. Additionally, we propose using NMIA [4]. NMIA defines a bag can consisting of multiple sub-bags, which contain the instances themselves. This serves to further stratify into clusters or regions, and accurately represent the

arrangement of the scattered data, where tiles belong to particular tissue areas and these themselves to the WSI. A bag-of-bags for a WSI $\mathbf{H}_{\text{WSI}}$ contains a set of inner-bags, or regions, $\mathbf{H}_{\text{REG},k}$:

$$\mathbf{H}_{\text{WSI}} = \left\{ \mathbf{H}_{\text{REG},k}, \forall k = 1, ..., K \right\} \tag{7}$$

where the number of inner-bags $K$ varies between different WSI. Ultimately, $\mathbf{H}_{\text{REG},k}$ contain instance-level representations $\mathbf{h}_{\text{TILE},l}$ of tiles located within the physical region.

For this project, we explored three configurations: using image data, using clinicopathological data and fusing both. For image data, we applied a weakly supervised architecture. For clinical data, we sorted it into a 1-dimensional vector $\mathbf{h}_{var}$ and fed it directly to the patient classifier $\Psi_\rho$. The clinical vector $\mathbf{h}_{var}$ is created by assigning integer values to clinical parameters based on their real-world values (e.g., 0 for non-smoker, 1 for smoker), see Supporting Information S1 Table annex. The length of this vector corresponds to the number of variables used. As for the combination of both, we used a weakly supervised architecture for generating the patient embedding representation and concatenated the clinical features to said embedding as $\mathbf{h}_{cli} = \left[ \mathbf{A} \cdot \mathbf{H}, \mathbf{h}_{var} \right]$.

### Deep learning pipeline method for prognostics

Our method integrates an automatic ROI tile extraction algorithm, a self-supervised-trained CNN-based feature extraction module, and a weakly supervised aggregation of visual features for prognostic outcome prediction, as displayed in Fig 1. Ultimately, these individual blocks of the pipeline sequentially process raw data extracted from NMIBC WSI in order to produce a prediction.

In the following sections, we will present the results obtained using our proposed method. We will discuss the performance metrics, comparative analysis with baseline models, and insights gained from the attention mechanisms used in the MIL framework, annotations and clinicopathological data.

### Experimental setup

In this section, we present the carefully designed experiments. First, to evaluate the proposed pipeline structure, we present experiments on artificial data as well as an application with region based labels. From there on, we consider two prognostic applications: BCG-RP and recurrence prediction. Hyperparameters, like the choice of optimizer, loss, and others are identical in all experiments. The results are presented in terms of AUC as our primary metric due to its robustness in evaluating model performance across different threshold settings. However, additional metrics such as sensitivity and specificity provide a more comprehensive evaluation for imbalanced or clinically sensitive applications. The code is available at https://github.com/Biomedical-Data-Analysis-Laboratory/HistoPrognostics.

**ROI extraction.** The fully automatic tissue segmentation was performed according to [40] for recurrence $S_{\text{SUH}}$ and [41] for BCG treatment $S_{\text{EMC}}$. Different regions were extracted as explained in section Automatic Region of Interest Segmentation.

**Contrastive feature representations.** We define a temperature parameter to 0.07 for the calculation of the loss for contrastive learning, as explained in section Feature Extraction via Contrastive Learning. We experiment with different CNN backbones; VGG16, DenseNet121 and ResNet18, with initial weights $\theta_I$, from pretraining on ImageNet [47]. For the supervised approaches, labels of grading and presence of TILs were used as diagnostic factors. In order to weigh the impact of the training size, we also run experiments of the unsupervised variant with larger, but limited, training samples. With respect to multi-task learning of unsupervised

contrastive and cross entropy classification, we set the parameters $\alpha_c$ and $\alpha_{ce}$ to 1.0 and 0.5, respectively. Adam was set as optimizer with a learning rate of 1e-4, a batch size of 128, and a total of fixed 10 epochs. The augmentations applied consisted of flip, flop, rotation, affine transformations, and color jittering.

**Prognosis classification.**   Bladder cancer recurrence is indeed regarded as a manifestation of treatment failure, as it indicates that the initial treatment did not effectively eradicate all cancer cells. Building upon this rationale, we will employ the data set $S_{EMC}$, see Table 1, in our decision-making process regarding the selection of feature extractors, contrastive loss functions, ROI selection, and magnification levels for both prognostic applications. This choice derives from the larger number of patients in the dataset, being a more representative sample of the population under study. Focal Tversky loss (FTL) is employed [48]. FTL employs two parameters, denoted as $\alpha_l$ and $\gamma_l$, which allow for adjusting the focus on different classes and handling the difficulty of training examples, respectfully. We also set an early stopping criteria of 30 epochs based on the AUC score on the validation set. We also implement a 5-runs Montecarlo with a 5% dropout for sampling purposes. A grid hyperparameter search is done to find the optimal values for the given task. The search includes bag sampling $n_b$, learning rate $lr$, optimizer $opt$, dropout rate $d_r$, number of neurons in the classifier $n_{\Theta_\rho}$ and attention mechanism $n_{att}$, loss functions parameters $\alpha_l$ and $\gamma_l$. The list of hyperparameters with their corresponding possible values and the resulting choice can be seen in Table 2:

## Preliminary experimentation

The three-step pipeline is convoluted, and prognostic labels are weak, typically involving one label per patient. To evaluate the pipeline, we conduct experiments in more controlled settings. Firstly, we aim to assess the pipeline using synthetic data generated from distributions with varying degrees of overlap. Thereafter, we evaluate the pipeline on actual WSI data, focusing on a diagnostic task with strong region-based labels, specifically the detection of regions containing lymphocytes.

### Effects of different data distributions

We define three different matching distributions pairs, which can be described as distributions with minor, partial overlap, or significant overlap. The degree of overlap is tuned using the mean $\mu$ and standard deviation $\sigma$ parameters. For all experiments, we define $\mathcal{P}_{3,2.5}$ as our class 0. For the matching distribution of class 1, we use $\mathcal{P}_{-3,1}$, $\mathcal{P}_{0,2}$ and $\mathcal{P}_{2,1.5}$ for minor, partial and significant overlap, respectively. A total of 150 bags are created, balanced among two classes, with a 90, 30, 30 split for train, validation and test, respectively. The number of

**Table 2. Results of hyperparameter search for optimizing predictive performance.**

| Hyperparameter | List of values |
| --- | --- |
| $lr$ | $10^{-1}, 10^{-2}, 10^{-3}, 10^{-4}$ |
| $opt$ | SGD, Adam |
| $n_b$ | 4, 16, 64, 256, $L$ |
| $d_r$ | 0.1, 0.2, 0.3, 0.4, 0.5 |
| $\alpha_l$ | 0.0, 0.3, 0.6, 0.9 |
| $\gamma_l$ | 0.5, 1, 2 |
| $n_{\Theta_\rho}$ | 128, 512, 1024, 4096 |
| $n_{att}$ | 128, 512, 1024, 4096 |
| **Hyperparameter choice** | $lr = 10^{-2}$, $opt =$ SGD, $n_b = 64$ $d_r = 0.2$, $\alpha_l = 0.9$, $\gamma_l = 2.0$, $n_{\Theta_\rho} = \{4096, 2048\}$, $n_{att} = 4096$ |

instances per bag is variable $N \in [3000, 7000]$, without replacement. The number of positive instances is limited to be less than half.

Results indicate that it is uncomplicated to discern between $\mathcal{P}_{-3,1}$ and $\mathcal{P}_{0,2}$ distributions, as shown in Table 3. However, we observe that for an overlapping distribution $\mathcal{P}_{2,1.5}$ the learning breaks down. We wanted to go further into finding the breaking point for the partial overlap distribution $\mathcal{P}_{0,2}$, and as such, we tried different parameters regarding the bag label balance in the train set and the representation of positive samples in bags. We look into a percentage of positive bags of 25 and 40%, as well as the number of positive samples in the range of 0 to 50%, and 0 to 75%. Results reveal the significance of class imbalance in bag classification, and the representation of positive and negative samples within the bags. The classification of highly imbalanced bag datasets can be challenging. While augmenting the number of positive instances can enhance the instance classification performance, it is insufficient to achieve satisfactory bag classification results. The performance naturally improves as the bag distribution becomes more balanced, but it is not enough unless the positive class is over-represented.

## Detection of lymphocytes

We evaluated the performance of our proposed solution in a diagnostic task using the same WSI as those used in $S_{\text{EMC}}$. Specifically, we defined a problem of detecting the presence of tumor-infiltrating lymphocytes (TILs), immune cells which have been associated with improved patient outcomes [6]. Tiles of size $256 \times 256$ were extracted from annotated lamina propria areas, with and without TILs, at 40x magnification. As a pre-processing step, tiles were normalized and resized to $224 \times 224$. Augmentations performed consisted of rotation and jittering operations. We define two experiments for different percentages of tiles with stromal TILs. The first one, $MIL_{1+}$, follows the standard MIL convention of detecting at least one tile with TILs. As for the second one, $MIL_{t+}$, we define a threshold $t$ for which more than 50% of the tiles must contain TILs.

We employed $MIL_{1+}$ to determine the optimal configuration for feature extraction. The results are presented in Table 4. Pre-trained features from natural images indeed capture relevant features for histopathological classification. However, the unsupervised approach $\theta_C$ exhibits superior performance in bag-level classification. $MIL_{t+}$ is therefore only tested with $G_{\theta_C}$, and this provides the best result with an AUC of 1.0. This might be due to the fact that labels are noisy, as many of the tiles annotated TIL-free actually contain some, even if the count is low. Using $MIL_{t+}$ makes it easier for the model to find a threshold $t$ than an absolute presence or absence given the noisy weak labels. Consistent with the observations on synthetic data, bags containing a substantial number of positive instances exhibit superior performance. This implies that accurate predictions rely on the existence of several positive instances within a bag, with less emphasis on the presence of positive instance outliers.

## Prognostic experiments

In this section, we utilize the proposed three-step pipeline for two prognostic applications: BCG response prediction on the $S_{\text{EMC}}$ dataset, and recurrence prediction on the $S_{\text{SUH}}$ dataset.

**Table 3. Prediction probabilities for discerning $\mathcal{P}_{3,2.5}$.**

| Class 1 | AUC |
|---|---|
| $\mathcal{P}_{-3,1}$ | **1.000** |
| $\mathcal{P}_{0,2}$ | **1.000** |
| $\mathcal{P}_{2,1.5}$ | 0.500 |

**Table 4. TIL detection comparison for different feature extractors.**

| MIL | Weights $\theta$ | AUC |
|---|---|---|
| $MIL_{1+}$ | $\theta_I$ | 0.833 |
| | $\theta_{CE}$ | 0.938 |
| | $\theta_{SC}$ | 0.909 |
| | $\theta_C$ | 0.940 |
| | $\theta_{MULTI}$ | 0.938 |
| $MIL_{t+}$ | $\theta_C$ | **1.000** |

Given the numerous choices for contrastive learning loss, backbone, ROI, and magnification levels, coupled with the computational intensity of learning and inference on gigapixel images, we adopt a systematic experiment approach, as illustrated in Fig 4. Therefore, we focus on testing one factor at a time and restrict further testing to the most promising results. In order to identify the optimal choices, we couple an AbMIL classification module to assess the resulting classification performance, using the validation subset. In subsections Feature Extraction and Contrastive Learning to On the Importance of Manual Annotations, we use BCG-RP with $S_{EMC}$ as the reference task, while recurrence prediction with $S_{SUH}$ is discussed in subsection On the Importance of Manual Annotations. Finally, subsection Attention-guided Interpretability discusses the interpretability of trained models.

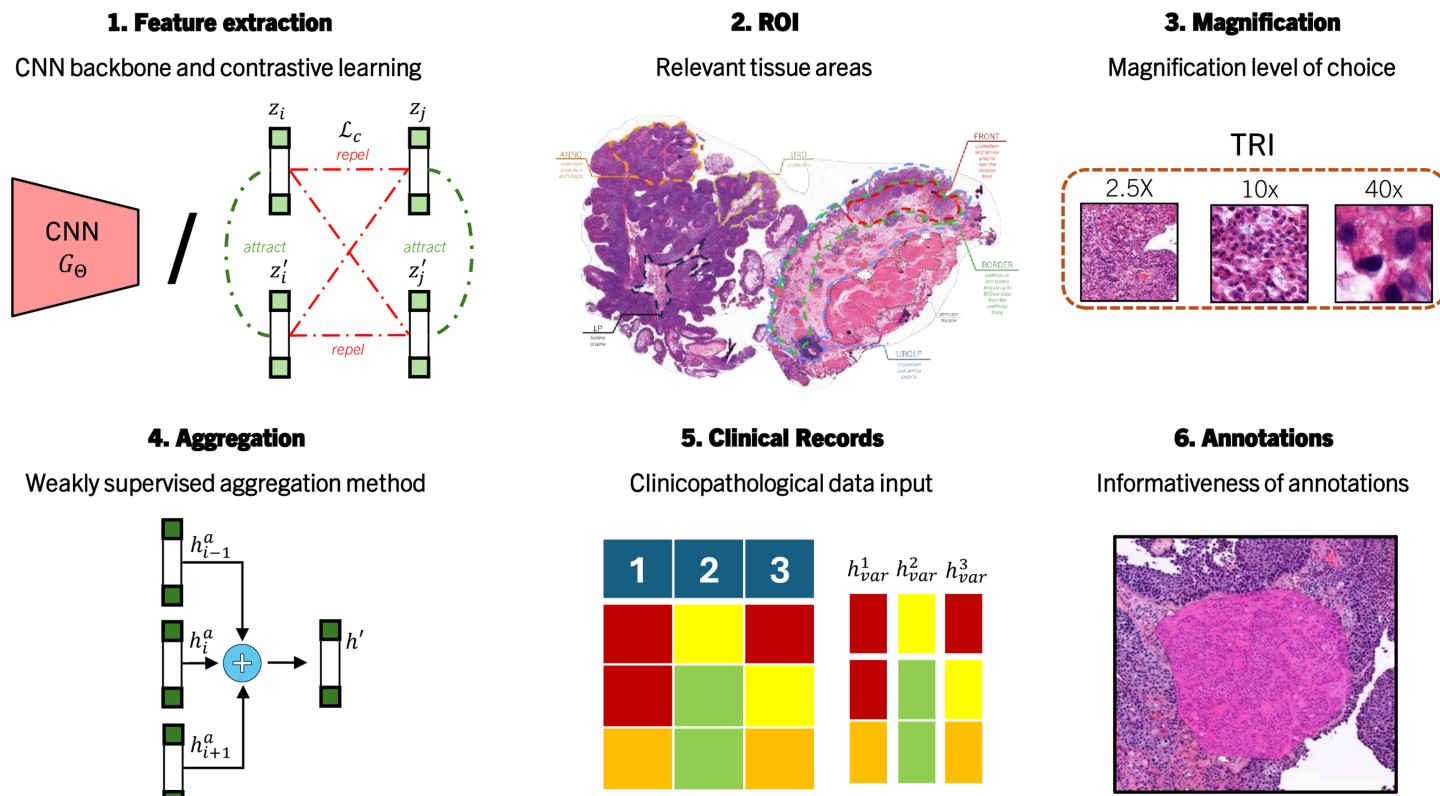

**Fig 4. Experimental diagram of prognostic experiments on NMIBC WSI outcome prediction.** Following a systematic experimental approach, diverse design parameters are progressively set based on validation set results. The order of the experiments carried out follows the order displayed.

### Feature extraction and contrastive learning

We aim to select a CNN backbone of preference for the feature extractor $G_\theta$, referring to the second step in the pipeline. The most commonly used backbones in CPATH literature are DenseNet, ResNet and VGG [7]. To ensure a fair performance comparison in a contrastive learning approach, we will incorporate the use of different labels, which will either be unsupervised, supervised contrastive or multi-task learning. In this experiment, we constrict the ROI to be $D_{\text{URO}}^{20x}$, with patches extracted at 20x. The urothelium tissue type is widely recognized as highly informative, and the choice of 20x magnification strikes a balance between capturing morphological structure context and preserving cellular details.

Results for classification can be found in Table 5. DenseNet121 offers promising results in terms of AUC performance for classification, without compromising computational efficiency. Therefore, we will use DenseNet121 as the backbone for the remaining experiments. The best performing contrastive learning strategy is unsupervised $\mathcal{L}_c$. Hence, we will proceed to use the frozen weights from the unsupervised method $\theta_C$.

### Region of interest selection

After determining the CNN backbone and contrastive learning strategy, the next step involves identifying the ROI with the best discriminative capability for the prognostic task. As highlighted in Fig 2, the ROIs are found from either manual annotations or from the output of the automated tissue segmentation model. From the automatically segmented regions, we obtain $D_{\text{URO}}^{20x}$, $D_{\text{LP}}^{20x}$, $D_{\text{UROLP}}^{20x}$, $D_{\text{BORDER}}^{20x}$, $D_{\text{FRONT}}^{20x}$, while $D_{\text{ANNO}}^{20x}$ is generated from the annotated set.

A comparison is shown in Fig 5. The comparison underscores $D_{\text{UROLP}}^{20x}$'s advantage, showcasing its consistent and balanced classification performance with an average AUC score of 0.728. That said, $D_{\text{FRONT}}^{20x}$ has the lowest variance across runs, reaching the most consistent results. $D_{\text{ANNO}}^{20x}$ demonstrates the second-highest average AUC score, but the highest variance. This implies a promising characteristic for future development of diagnostic models. Emulating a pathologist sense of expertise, a model could automatically identify diagnostically relevant areas and generate annotations. Then, these generated annotations could be used for further enhancing feature discrimination for subsequent prognostic models. That said, in our investigation, only a limited set of tiles is annotated. This results in a mismatch in training size between datasets automatically generated and annotations. Moreover, an independent system cannot rely on expert input during the inference stage. Moving forward, we take all three aforementioned ROIs for determining the optimal magnification input choice.

### Magnification level choice

We aim to assess the influence of varying magnification levels on the performance, testing mono-scale magnification levels of 10x, 20x and multi-scale TRI (2.5x, 10x, 40x). The results are shown in Table 6, and allows us to assess the model's accuracy in capturing both broader

**Table 5. Validation AUC scores for $D_{\text{URO}}^{20x}$ BCG-RP.**

| Weights $\theta$ | DenseNet121 | ResNet18 | VGG16 |
|---|---|---|---|
| $\theta_I$ | 0.576(0.029) | 0.474(0.009) | 0.549(0.005) |
| $\theta_C$ | **0.672(0.032)** | 0.628(0.017) | 0.506(0.022) |
| $\theta_{SC}$ | 0.521(0.054) | 0.568(0.009) | 0.480(0.036) |
| $\theta_{MULTI}$ | 0.515(0.031) | 0.506(0.042) | 0.434(0.048) |

*The results show the mean and standard deviation over 5 runs.*

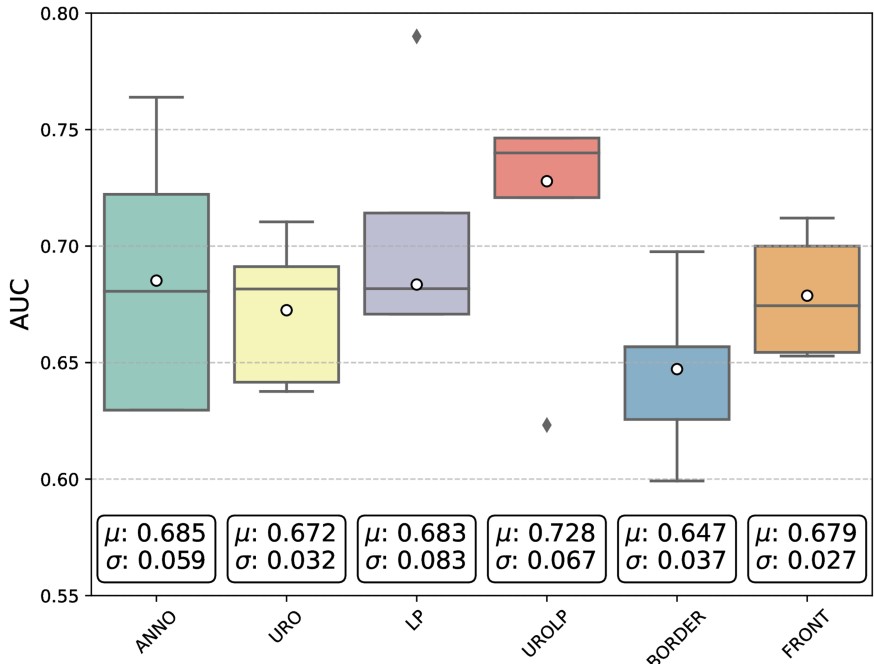

**Fig 5. Box plot illustrating AUC performance variation across validation sets with different ROIs for 20x magnification.** Notably, $D_{UROLP}^{20x}$ emerges as the top performer amongst the ROIs. White dots represent the average value $\mu$, and black diamonds represent outliers. The results show the mean $\mu$ and standard deviation $\sigma$ over 5 runs.

**Table 6. Validation AUC scores for BCG-RP at different ROIs and magnification levels.**

| Magnification | UROLP | FRONT | ANNO* |
|---|---|---|---|
| **10x** | 0.621(0.021) | 0.647(0.033) | **0.790(0.113)** |
| **20x** | **0.728(0.067)** | 0.679(0.027) | 0.685(0.059) |
| **TRI** | 0.649(0.014) | 0.621(0.038) | 0.615(0.029) |

*The results show the mean and standard deviation over 5 runs.*
*\* Not all train and validation WSI were annotated.*

tissue context and finer cellular details, as well as contextual neighbouring regions as per the multi-scale input.

Utilizing a fixed magnification level throughout the image consistently yields AUC levels that hover around 0.7 at 20x magnification, the highest for the automatic ROI $D_{UROLP}^{20x}$. However, it's worth noting that the predictive performance for 10x is worse. For comparison, $D_{ANNO}^{10x}$ obtained on average higher AUC values, yet, constrained to expert input and lack of a complete set of annotations. Despite of its capacity to capture structural intricacies, the multi-scale model TRI stands with lower performance. TRI performs worse possibly because the larger input requires more training data for generalization.

## Weakly supervised aggregation for treatment outcome prediction

We conduct a comparison between five aggregation techniques in weakly supervised learning. These include majority voting, max, mean, AbMIL and NMIA. These techniques are evaluated to determine their effectiveness in combining and summarizing information from multiple instances.

In Table 7, we present the obtained results on the test set. Among these techniques, those involving majority voting, mean and max aggregation, which do not include in-built attention mechanisms, demonstrate less promising outcomes. In contrast, the utilization of AbMIL leads to a notable performance improvement. Moreover, when examining the scattered tissue regions across the WSI using the nested multiple instance method, NMIA, we observe the most significant performance enhancement across all techniques with an AUC of 0.678. We observe a bias gap between the validation and test results, which hint a lack of generalizability or distinct distributions in between sets. Additionally, it is important to emphasize that models exhibiting this level of performance currently demonstrate limited reliability and remain far from being suitable for real-world clinical implementation.

## Fusing image and clinicopathological data

By fusing clinicopathological and image data with the aim of complementing each other, as explained in subsection Prognostic Outcome Classification via Multiple Instance Learning, deep learning models should gain a comprehensive understanding of patient conditions. However, it is imperative to acknowledge the potential limitations and challenges inherent in each form of data. While images offer visual cues that might be ambiguous without clinical context, reports may lack the granularity and specificity present in visual data. Clinicopathological data is often compiled from various sources, including physician notes, laboratory test results, and imaging, potentially introducing noise and variability. Conversely, WSI may also contain inherent variability due to factors such as staining variations or tissue preparation techniques.

In our experiments, unexpectedly, the findings displayed in Table 7 indicate that histological features may offer more pertinent information for predicting patient outcomes compared to either clinical data alone or their fusion. This finding raises questions about the traditional reliance on clinical data and highlights the potential of histopathological images as a standalone predictor for improved prognostic accuracy. It is noteworthy that the most valuable clinical data is derived from the manual visual examination of WSIs (e.g., grade, stage), leading to information redundancy and potential overfitting when combining the model's features with the clinical data vector. Another possible explanation for unexpected results might stem from the lack of normalization when using the clinical data, which may introduce variability and bias into the model, negatively affecting the learning. Nevertheless, moving forward, exploring different fusion methods for integrating clinicopathological and image data could

**Table 7. Test AUC performance for $D_{\text{UROLP}}^{20x}$ for BCG-RP and recurrence prediction.**

| Method | BCG | | | Recurrence | | |
|---|---|---|---|---|---|---|
| | Best Run | $\mu(\sigma)$ | Montecarlo | Best Run | $\mu(\sigma)$ | Montecarlo |
| Maj. Voting | 0.500 | 0.497(0.003) | 0.440(0.075) | 0.421 | 0.420(0.001) | 0.420(0.001) |
| Mean | 0.427 | 0.417(0.007) | 0.446(0.033) | 0.615 | **0.583(0.023)** | 0.613(0.015) |
| Max | 0.520 | **0.499(0.011)** | 0.487(0.007) | 0.555 | 0.517(0.047) | 0.544(0.008) |
| AbMIL | 0.549 | 0.434(0.077) | 0.548(0.007) | 0.592 | 0.500(0.064) | 0.593(0.002) |
| NMIA | **0.678** | 0.486(0.111) | **0.612(0.009)** | **0.721** | 0.580(0.079) | **0.722(0.003)** |
| Clinical | 0.501 | 0.471(0.024) | 0.498(0.017) | - | - | - |
| Clinical + AbMIL | 0.502 | 0.456(0.037) | 0.491(0.006) | - | - | - |
| Clinical + NMIA | 0.475 | 0.450(0.036) | 0.454(0.008) | - | - | - |

*The results show the mean and standard deviation over 5 runs.*

yield insights into the optimal approach for maximizing predictive performance. Moreover, multimodal fusion models might be a more suitable solution for fusing different data modalities than our proposed framework. Finally, incorporating additional clinical parameters may provide a more comprehensive understanding of the status of the disease.

## On the importance of manual annotations

At the inference stage, a fully-independent system cannot be conditioned by manually annotated regions. Even at the training stage, we have observed the challenges in obtaining annotations for all WSI due to the labor-intensive nature of the process. For a prognostic task, the relevance of manual annotations remains uncertain. To explore this aspect, we conducted an experiment where the model was trained on annotated regions ANNO, but tested on automatically segmented regions AUTO, and vice versa. This was compared to a fully automated system for both training and inference. Table 8 shows the results, where the ROI-column indicates which ROI definition was used for the AUTO set. For comparison, using ANNO for both train and test gives a performance of 0.441(0.031). With one exception, the models perform better after using AUTO-generated ROIs in the training and ANNO for testing, which we interpret as the models benefiting from the larger dataset that is available when we can include non-annotated data.

## Recurrence prediction

Following the predefined settings for ROI, magnification, and feature extraction used in the BCG application, we apply the same configuration for recurrence prediction. The results are presented in Table 7. We observe that the utilization of mean aggregation demonstrated the highest average predictive performance. Nevertheless, NMIA exhibited the most substantial performance enhancement among all the techniques studied, with an AUC of 0.721. This underscores the effectiveness of incorporating nested attention mechanisms for the accurate identification of crucial patterns within tissue regions. However, as previously discussed in the context of BCG-RP, developing models based on a single private cohort limits generalizability, particularly when the predictive performance remains insufficient for clinical adoption.

## Attention-guided interpretability

The attention scores obtained at the inference stage can be useful for interpretability reasons and can give knowledge on what the model considers relevant for a given prediction. Instances with higher attention scores often correspond to pivotal regions pertinent to the predicted label, aiding in both model validation and reasoning. Attention scores aid pathologists in understanding model's rationale, highlighting areas of interest. This interpretability

**Table 8. Comparing AUC performance for distinct training and test datasets, for $S_{EMC}$ 20x magnification.**

| ROI (Train/Test) | Architecture | URO | LP | UROLP | BORDER | FRONT |
|---|---|---|---|---|---|---|
| ANNO / AUTO | AbMIL | 0.489(0.032) | 0.446(0.055) | **0.528(0.026)** | 0.463(0.054) | 0.408(0.025) |
| | NMIA | 0.543(0.022) | 0.476(0.054) | 0.475(0.032) | 0.468(0.054) | 0.503(0.027) |
| AUTO / ANNO | AbMIL | **0.611(0.067)** | 0.468(0.144) | 0.508(0.090) | **0.600(0.094)** | **0.663(0.098)** |
| | NMIA | 0.486(0.119) | **0.601(0.082)** | 0.519(0.054) | 0.525(0.054) | 0.531(0.073) |
| AUTO / AUTO | AbMIL | 0.492(0.065) | 0.452(0.060) | 0.434(0.077) | 0.534(0.036) | 0.484(0.066) |
| | NMIA | 0.535(0.046) | 0.513(0.043) | 0.486(0.111) | 0.512(0.053) | 0.428(0.028) |

*The results show the mean and standard deviation over 5 runs.*

is valuable for either positive or negative predictions, facilitating validation and refinement of the model's decisions.

An example of attention score heatmap is visualized in Fig 6. Pathologists often use tissue punching to select clinically relevant regions for subsequent analysis in tissue microarrays. While the model consistently allocates its highest attention to punched areas, it is crucial to acknowledge that diagnostically relevant information may extend beyond the punched areas. Nevertheless, the alignment between the model's attention focus and the clinical practice of region selection through punching is an encouraging and promising finding.

The heatmap in Fig 7 highlights the regions where the model assigns high attention scores in comparison to ROIs annotated by pathologists, based on clinical experience and histopathological criteria. The strong correlation between the model's high attention areas and the pathologists' annotations suggests that the model effectively identifies clinically relevant regions, supporting its use in diagnostic processes. By pinpointing critical regions, the model assists pathologists in focusing on the most diagnostically significant areas of a slide, enhancing both its transparency and practical utility in a clinical setting. This is particularly valuable in cases of NMIBC with extensive tissue samples, where manual review can be time-consuming. Comparing the attention disparity between classes, unlike the clear distinction seen at the region level, where BCG-R areas were prominently highlighted, the comparison between BCG-R and BCG-NR at the patch level does not reveal significant differences. However, we noted that, for both classes, the model consistently prioritizes areas annotated as cancerous and invasive over those marked for grading. This observation is encouraging, as advanced stages of the disease are typically associated with poorer outcomes.

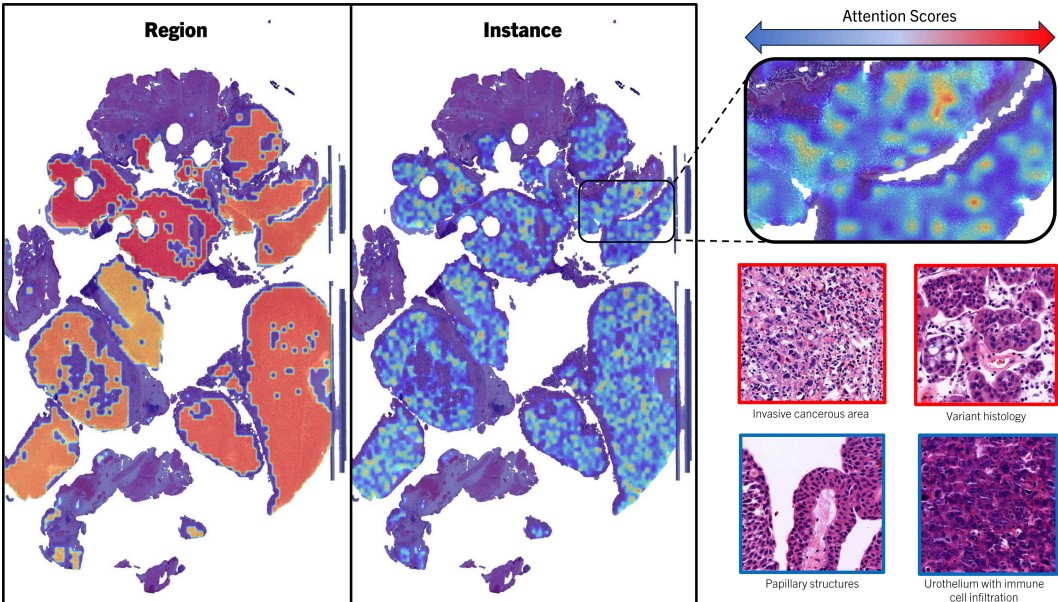

**Fig 6. Heatmap illustrating attention scores over a BCG-NR WSI from $S_{EMC}$.** The heatmap provides insights into the ROIs where the attention is concentrated within the WSI, facilitating a better understanding of prediction dynamics and highlighting areas of significance for clinical interpretation.

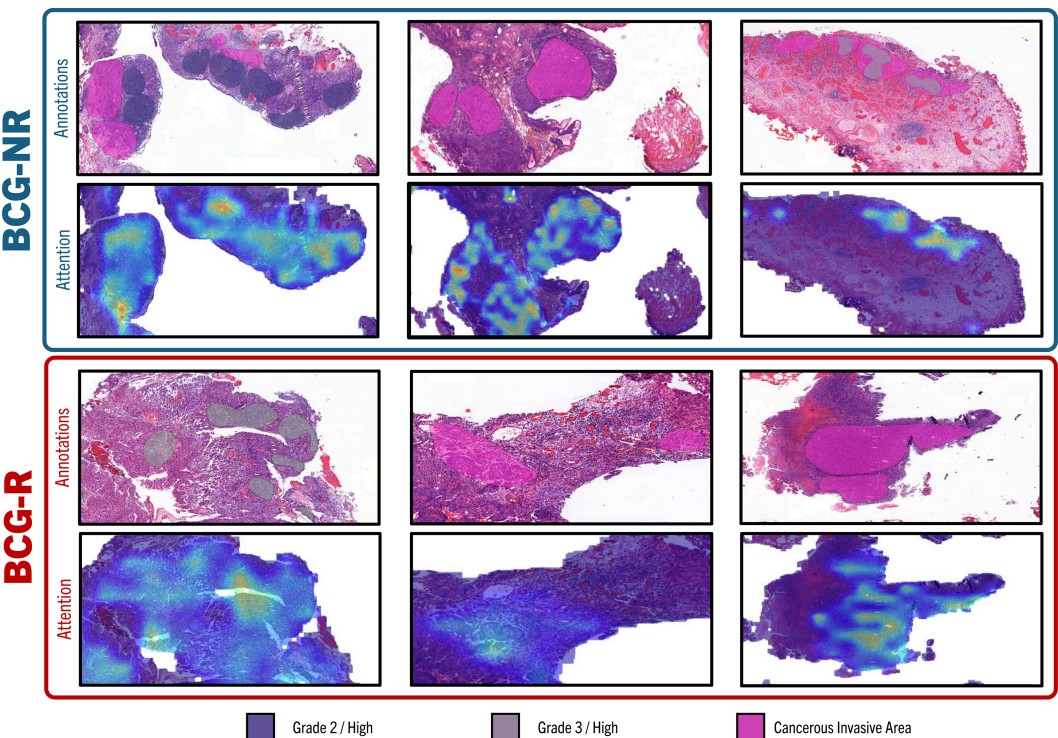

**Fig 7. Heatmap of attention scores over annotated areas.** Annotated areas by pathologists are overlaid for comparison, demonstrating the correlation between the model's predictions and clinical annotations.

## Conclusion

We propose a three-step automated pipeline for the challenging task of prognostic prediction in a real-world clinical setting, eliminating the need for manual annotations. This trend is expected to persist in the future of CPATH, given the numerous tasks and limited annotation resources available.

The process begins with automated ROI segmentation, where task-specific knowledge can be combined with an automatic tissue classifier to extract relevant ROI. Since prognostic labels are weak in nature, the second step employs contrastive learning to train the feature extractor. Finally, an attention-based nested multiple instance learning classifier, providing predictions and insights into the crucial regions of the image. The pipeline demonstrates exceptional performance on a simpler diagnostic task, achieving a perfect score with an AUC of 1.0 for detecting areas with tumor-infiltrating lymphocytes. Additionally, experiments on synthetic data confirm that the pipeline works perfectly when input distributions are distinct. However, performance drops when they become highly overlapping. In our study, we conducted a thorough pioneering investigation into the use of deep learning and histopathological images for prognostic prediction. We employ response to BCG treatment in patients with HR-NMIBC and recurrence in NMIBC patients as uses cases. The utilization of deep learning for BCG-RP using WSI is a first. The most promising outcomes reveal AUC values of 0.678 for BCG outcome prediction and 0.721 for recurrence prediction. While there is room for improvement, it is important to recognize that performance requirements must be substantially higher to meet the standards necessary for clinical implementation. Ultimately, achieving fully automated prognostics based solely on WSI remains a challenging task.

### Limitations and future research

Models trained on a single cohort may exhibit limitations in generalizing to images with diverse histological characteristics. An additional constraint is the development and evaluation of models using only in-house cohorts, which raises concerns about their generalizability for broader applications. To address these limitations, future research should focus on developing and validating the proposed methods with external cohorts to confirm their applicability across different populations. Furthermore, evaluating the statistical significance of predictions as a valid factor for risk stratification is crucial.

While histopathological features were prioritized in this study, routine practice often involves integrating additional data modalities, such as sequencing. Incorporating omics and immunohistochemistry into prognostic models will also be essential to align with the comprehensive information available to pathologists for disease management decisions. Additionally, examining how insights from diverse diagnostic models could be integrated to improve the quantitative analysis of disease risk is warranted. Future work should explore the implementation of multi-modal architectures or distinct fusion strategies to exploit and integrate these data modalities. Moreover, alternative architecture reliant on transformers or graphs could potentially raise the achieved performance.

A noted area for enhancement is the comparison with existing methods. Although the study has introduced innovative approaches, a broader comparison with current state-of-the-art techniques could provide valuable insights into their relative performance, potential advantages and areas for further improvement.

Finally, while our model demonstrates promising results, its current performance is only evaluated using the AUC metric, and does not yet meet the required reliability standards for treatment outcome prediction tasks. This comes as a result of the previously mentioned data and technical limitations and the inherent complexity of prognostic outcome prediction.

## Supporting information

**S1 Table. Clinicopathological data classes.**
(PDF)

## Author contributions

**Conceptualization:** Saul Fuster, Farbod Khoraminia, Trygve Eftestøl, Tahlita C. M. Zuiverloon, Kjersti Engan.

**Data curation:** Saul Fuster, Farbod Khoraminia, Umay Kiraz, Geert J. L. H. van Leenders, Tahlita C. M. Zuiverloon.

**Formal analysis:** Saul Fuster, Farbod Khoraminia, Trygve Eftestøl, Kjersti Engan.

**Funding acquisition:** Valery Naranjo, Emiel A. M. Janssen, Tahlita C. M. Zuiverloon, Kjersti Engan.

**Investigation:** Saul Fuster, Farbod Khoraminia, Julio Silva-Rodríguez, Trygve Eftestøl, Tahlita C. M. Zuiverloon, Kjersti Engan.

**Methodology:** Saul Fuster, Farbod Khoraminia, Julio Silva-Rodríguez, Trygve Eftestøl, Kjersti Engan.

**Project administration:** Valery Naranjo, Emiel A. M. Janssen, Tahlita C. M. Zuiverloon, Kjersti Engan.

**Resources:** Farbod Khoraminia, Emiel A. M. Janssen, Tahlita C. M. Zuiverloon.

**Software:** Saul Fuster, Julio Silva-Rodríguez.

**Supervision:** Julio Silva-Rodríguez, Geert J. L. H. van Leenders, Trygve Eftestøl, Valery Naranjo, Emiel A. M. Janssen, Kjersti Engan.

**Validation:** Saul Fuster, Farbod Khoraminia, Umay Kiraz, Geert J. L. H. van Leenders, Emiel A. M. Janssen, Tahlita C. M. Zuiverloon, Kjersti Engan.

**Visualization:** Saul Fuster, Farbod Khoraminia, Trygve Eftestøl, Kjersti Engan.

**Writing – original draft:** Saul Fuster, Trygve Eftestøl, Kjersti Engan.

**Writing – review & editing:** Saul Fuster, Farbod Khoraminia, Julio Silva-Rodríguez, Umay Kiraz, Geert J. L. H. van Leenders, Trygve Eftestøl, Valery Naranjo, Emiel A. M. Janssen, Tahlita C. M. Zuiverloon, Kjersti Engan.

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
