## [Decision Letter · Decision Letter 0]

5 Jul 2024

PDIG-D-24-00096

Self-Contrastive Weakly Supervised Learning Framework for Prognostic Prediction Using Whole Slide Images

PLOS Digital Health

Dear Dr. FUSTER,

Thank you for submitting your manuscript to PLOS Digital Health. After careful consideration, we feel that it has merit but does not fully meet PLOS Digital Health's publication criteria as it currently stands. Therefore, we invite you to submit a revised version of the manuscript that addresses the points raised during the review process.

Please submit your revised manuscript within 60 days Sep 03 2024 11:59PM. If you will need more time than this to complete your revisions, please reply to this message or contact the journal office at digitalhealth@plos.org. Please include the following items when submitting your revised manuscript:

We look forward to receiving your revised manuscript.

Kind regards,

Sivaramakrishnan Rajaraman, Ph.D.

Academic Editor

PLOS Digital Health

Journal Requirements:

1. We ask that a manuscript source file is provided at Revision. Please upload your manuscript file as a .doc, .docx, .rtf or .tex.

Additional Editor Comments (if provided):

In addition to addressing the reviewers' comments, I would recommend the authors to perform statistical significance analysis in the revised version, at every stage of the reported results to ensure that the observations did not occur by chance.

Reviewers' comments:

Reviewer's Responses to Questions

**Comments to the Author**

1. Does this manuscript meet PLOS Digital Health’s publication criteria? Is the manuscript technically sound, and do the data support the conclusions? The manuscript must describe methodologically and ethically rigorous research with conclusions that are appropriately drawn based on the data presented.

Reviewer #1: Yes

Reviewer #2: Yes

Reviewer #3: Yes

2. Has the statistical analysis been performed appropriately and rigorously?

Reviewer #1: I don't know

Reviewer #2: N/A

Reviewer #3: Yes

3. Have the authors made all data underlying the findings in their manuscript fully available (please refer to the Data Availability Statement at the start of the manuscript PDF file)?

Reviewer #1: Yes

Reviewer #2: No

Reviewer #3: Yes

4. Is the manuscript presented in an intelligible fashion and written in standard English?

PLOS Digital Health does not copyedit accepted manuscripts, so the language in submitted articles must be clear, correct, and unambiguous. Any typographical or grammatical errors should be corrected at revision, so please note any specific errors here.

Reviewer #1: Yes

Reviewer #2: Yes

Reviewer #3: Yes

5. Review Comments to the Author

Please use the space provided to explain your answers to the questions above. You may also include additional comments for the author, including concerns about dual publication, research ethics, or publication ethics. (Please upload your review as an attachment if it exceeds 20,000 characters)

Reviewer #1: I am grateful for the opportunity to review the study titled "Self-Contrastive Weakly Supervised Learning Framework for Prognostic Prediction Using Whole Slide Images" The authors have presented a prognostic model for predicting bladder cancer recurrence and treatment response based on tissue slide images. This research represents a significant contribution to the field of computational pathology. However, several points need addressing to enhance the clarity of the study and to address some limitations.

Minor

1. I suggest that the fourth line of the paragraph be augmented with a citation to a related paper. This citation would serve to add more context to the nature of weakly labeled data.

2. A comprehensive reorganization of the paper is warranted. The Introduction section could hold the Background & Related Work, Data Modalities, and Non-Supervised Learning Methods sections. Furthermore, the Dataset and Compliance with Ethical Standards should be relocated to the Method section. A reconstruction to facilitate a clearer demarcation between the Method section and the Results, will ultimately enhancing readability.

3. I recommend the inclusion of a flow diagram to illustrate the experimental workflow. This visual aid would provide clarity regarding the model optimization process undertaken.

4. In Table 1 it’s advisable to mention the abbreviation meanings in the caption instead of only mentioning them in the text.

Major

1. Several models showcased in Table 7 exhibited performance metrics comparable to random guess or slightly better. The absence of additional metrics precludes a comprehensive analysis of these outcomes and a detailed comparison between the models. The authors overall didn’t include a metric other than AUC for evaluating model performance although other metrics such as accuracy and precision are valuable especially for treatment prognosis which may be the basis for choosing a treatment.

2. At line 460 the clinical data fed to the model is ambiguous. The unexpected result raises questions about the nature and the form the clinical data where fed to the model.

Reviewer #2: While the paper presents a comprehensive approach to prognostic prediction using histopathological images and deep learning techniques, several potential issues should be addressed:

1. Limited Comparison with Existing Methods: The paper lacks a thorough comparison with existing prognostic prediction methods, especially those that integrate clinical and histopathological data. A more extensive comparison would provide a better context for evaluating the proposed approach's effectiveness.

2. Dataset Limitations: The paper should provide more information about the datasets used, including their size, diversity, and representativeness. Additionally, addressing any potential biases or limitations in the datasets would strengthen the validity of the results.

3. Validation and Generalization: The paper mentions validation results, but it's unclear how well the proposed method generalizes to new datasets or populations. Providing insights into the model's performance across different datasets or in real-world clinical settings would enhance the paper's applicability and reliability.

4. Interpretability and Clinical Relevance: While attention-guided interpretability is discussed, the paper could delve deeper into how the model's predictions align with clinical practice and whether the identified regions of interest are clinically meaningful. Providing examples of how the model's predictions could inform clinical decision-making would strengthen the paper's relevance to healthcare practitioners.

5. Future Directions: While the paper discusses potential improvements and future directions briefly, a more detailed discussion on the limitations of the proposed approach and avenues for future research would provide valuable insights for the research community.

6. Contradiction: One potential contradictory idea in the paper is the finding that histological features may offer more pertinent information for predicting patient outcomes compared to either clinical data alone or their fusion. This contradicts the common practice of integrating clinical and histopathological data to improve prognostic accuracy. Typically, the medical field emphasizes the importance of considering both clinical and histopathological data for accurate prognosis. However, the paper suggests that histopathological images alone may be sufficient for predicting patient outcomes with higher accuracy. This finding challenges the traditional approach to prognostic prediction and raises questions about the relative importance of different types of data in healthcare decision-making. The paper needs to acknowledge this potential contradiction and provide a thorough discussion of the implications. This could include exploring possible reasons for the discrepancy, such as differences in data quality or the specific characteristics of the study population. Additionally, further research could be recommended to validate the findings and determine the optimal approach for integrating clinical and histopathological data in prognostic prediction.

Reviewer #3: The manuscript presents a novel deep learning framework for prognostic prediction from histopathological whole slide images (WSIs) in urinary bladder cancer. It combines tissue segmentation, contrastive learning for feature extraction, and nested multiple instance learning (NMIA) for classification, achieving AUCs of 0.721 and 0.678 for recurrence and treatment outcome prediction, respectively. While innovative, the manuscript would benefit from more detailed comparisons with existing deep learning methods (PMID: 36268091)and the reasons for choosing NMIA as well as comparison with other prognostic studies ((PMID: 18367107). Also consider have abbreviation list.

6. PLOS authors have the option to publish the peer review history of their article (what does this mean?). If published, this will include your full peer review and any attached files.

**Do you want your identity to be public for this peer review?** For information about this choice, including consent withdrawal, please see our Privacy Policy.

Reviewer #1: No

Reviewer #2: No

Reviewer #3: No

---

## [Decision Letter · Decision Letter 1]

3 Jun 2025

PDIG-D-24-00096R1Self-Contrastive Weakly Supervised Learning Framework for Prognostic Prediction Using Whole Slide ImagesPLOS Digital Health Dear Dr. FUSTER, Thank you for submitting your manuscript to PLOS Digital Health. After careful consideration, we feel that it has merit but does not fully meet PLOS Digital Health's publication criteria as it currently stands. Therefore, we invite you to submit a revised version of the manuscript that addresses the points raised during the review process. Please submit your revised manuscript within 30 days Jul 03 2025 11:59PM. If you will need more time than this to complete your revisions, please reply to this message or contact the journal office at digitalhealth@plos.org. Please include the following items when submitting your revised manuscript:* A rebuttal letter that responds to each point raised by the editor and reviewer(s). You should upload this letter as a separate file labeled 'Response to Reviewers'. This file does not need to include responses to any formatting updates and technical items listed in the 'Journal Requirements' section below.* A marked-up copy of your manuscript that highlights changes made to the original version. You should upload this as a separate file labeled 'Revised Manuscript with Track Changes'.* An unmarked version of your revised paper without tracked changes. You should upload this as a separate file labeled 'Manuscript'. If you would like to make changes to your financial disclosure, competing interests statement, or data availability statement, please make these updates within the submission form at the time of resubmission. Guidelines for resubmitting your figure files are available below the reviewer comments at the end of this letter. We look forward to receiving your revised manuscript. Kind regards, Sebastián Andrés Cajas, M.s.C Image Processing And Computer VisionGuest EditorPLOS Digital Health Sebastián CajasGuest EditorPLOS Digital Health Leo Anthony CeliEditor-in-ChiefPLOS Digital Healthorcid.org/0000-0001-6712-6626**Additional Editor Comments (if provided):** While the manuscript presents a relevant and promising contribution, several key concerns remain partially addressed. We believe the paper can be accepted pending minor revisions, provided the following points are clearly and transparently reflected throughout:

- AUC Limitation: Acknowledge that AUC alone is insufficient to fully evaluate performance, especially in imbalanced or clinically sensitive contexts where metrics like precision or sensitivity are critical.

- Predictive Performance: Clearly discuss why the model’s performance may not yet meet standards for clinical use - comparing table 7 and validation results, particularly in treatment outcome tasks requiring higher reliability.

- External Validation: Explicitly recognize the lack of external validation as a methodological limitation affecting generalizability.

- Single Dataset Use: Emphasize that the study was conducted on a single dataset and ensure this constraint is clearly reflected across the abstract, discussion, conclusions, and tone.

- Future Work: Suggest directions to improve multimodal fusion, such as attention mechanisms, GNNs, or late fusion strategies.

These omissions affect the robustness and generalizability of the findings. It is therefore essential that the limitations of the study are clearly and consistently communicated not only in the abstract, but throughout the manuscript, including the discussion, conclusions, and overall tone. Without this, readers may overinterpret the applicability of the results. That said, we believe the paper can be accepted pending these adjustments, provided that the manuscript clearly acknowledges where it falls short, presents these limitations with transparency, and frames the contribution as a foundation for future work rather than a broadly validated solution.**Reviewers' Comments:** Reviewer's Responses to Questions

**Comments to the Author**

1. If the authors have adequately addressed your comments raised in a previous round of review and you feel that this manuscript is now acceptable for publication, you may indicate that here to bypass the “Comments to the Author” section, enter your conflict of interest statement in the “Confidential to Editor” section, and submit your "Accept" recommendation.

Reviewer #1: All comments have been addressed

Reviewer #4: All comments have been addressed

2. Does this manuscript meet PLOS Digital Health’s publication criteria? Is the manuscript technically sound, and do the data support the conclusions? The manuscript must describe methodologically and ethically rigorous research with conclusions that are appropriately drawn based on the data presented.

Reviewer #1: Yes

Reviewer #4: Yes

3. Has the statistical analysis been performed appropriately and rigorously?

Reviewer #1: I don't know

Reviewer #4: Yes

4. Have the authors made all data underlying the findings in their manuscript fully available (please refer to the Data Availability Statement at the start of the manuscript PDF file)?

Reviewer #1: No

Reviewer #4: Yes

5. Is the manuscript presented in an intelligible fashion and written in standard English?

Reviewer #1: Yes

Reviewer #4: Yes

6. Review Comments to the Author

Reviewer #1: The concerns have been adequately addressed. However, please note that in line 197, 'an' should be replaced with 'a'.

Reviewer #4: The authors have successfully addressed all my comments. I have no further suggestions. Based on my evaluation, the paper is now suitable for publication in PLOS Digital Health.

7. PLOS authors have the option to publish the peer review history of their article (what does this mean?). If published, this will include your full peer review and any attached files.

**Do you want your identity to be public for this peer review?** For information about this choice, including consent withdrawal, please see our Privacy Policy.

Reviewer #1: No

Reviewer #4: **Yes: **Sameena Naaz

---

## [Editor Report · Decision Letter 2]

20 Jul 2025

Self-Contrastive Weakly Supervised Learning Framework for Prognostic Prediction Using Whole Slide Images

PDIG-D-24-00096R2

Mr. SAUL FUSTER

We are pleased to inform you that your manuscript 'Self-Contrastive Weakly Supervised Learning Framework for Prognostic Prediction Using Whole Slide Images' has been provisionally accepted for publication in PLOS Digital Health.

Best regards,

Sebastián Andrés Cajas, M.s.C Image Processing And Computer Vision

Guest Editor

PLOS Digital Health